

# A DMA-train for precision measurement of sub 10-nm aerosol dynamics

Dominik Stolzenburg[1], Gerhard Steiner[1,2], and Paul M. Winkler[1]

[1]Faculty of Physics, University of Vienna, 1090 Vienna, Austria
[2]Institute for Ion and Applied Physics, University of Innsbruck, 6020 Innsbruck, Austria

*Correspondence to:* Dominik Stolzenburg (dominik.stolzenburg@univie.ac.at)

**Abstract.** Measurements of aerosol dynamics in the sub-10 nm size range are crucially important for quantifying the impact of new particle formation onto the global budget of cloud condensation nuclei. Here we present the development and characterization of a differential mobility analyzer - train (DMA-train), operating six DMAs in parallel for high time-resolution particle size-distribution measurements below 10 nm. The DMAs are operated at six different but fixed voltages and hence

sizes, together with six state-of-the-art condensation particle counters. Two Airmodus A 10 particle size magnifiers (PSM) are used for channels below 2.5 nm while sizes above 2.5 nm are detected by TSI 3776 butanol or TSI 3788 water based CPCs. We report the transfer functions and characteristics of six identical Grimm S-DMAs as well as the calibration of a butanol-based TSI model 3776 CPC, a water-based TSI model 3788 CPC and an Aimodus A10 PSM. We find cut-off diameters similar to those reported in the literature. The performance of the DMA-train is tested with a rapidly changing aerosol of a tungsten oxide

particle generator during warm-up. Additionally we report a measurement of new particle formation taken during a nucleation event in the CLOUD chamber experiment at CERN. We find that the DMA-train is able to bridge the gap between currently well-established measurement techniques in the cluster-particle transition regime, providing high time-resolution and accurate size information of neutral and charged particles even at atmospheric particle concentrations.

## 1 Introduction

Atmospheric aerosols still constitute the largest uncertainties in climate models through their ambiguous effects on the climate system (Carslaw et al., 2013). In addition to the direct effects of scattering and absorption on incoming solar radiation, higher aerosol number concentrations can increase the albedo of clouds (Ramanathan et al., 2001) and their lifetime (Albrecht, 1989) leading to significant indirect radiative forcing.

New particle formation from gaseous precursor vapours is frequently observed in the atmosphere (Kulmala et al., 2004).

Model simulations show that up to 50 % of the global budget of cloud condensation nuclei (CCN) might originate from new particle formation (Spracklen et al., 2008; Merikanto et al., 2009). However, these numbers strongly depend on the dynamics of the newly formed aerosol (Weber et al., 1997). Nucleation occurring at the critical cluster size between 1-2 nm is often followed by growth (Venzac et al., 2008; Riccobono et al., 2012) up to sizes of around 50-100 nm, where the particles can act as CCN (Lihavainen et al., 2003; Kerminen et al., 2005; Kuang et al., 2009). In competition with the growth by condensation



is the probability of the newly formed particles to get lost to bigger pre-existing aerosol. However, the driving mechanisms responsible for the aerosol growth are still largely unknown and require detailed studies of aerosol dynamics especially in the crucial sub-10 nm size range where coagulation losses are the highest.

Electrical mobility analysis is widely used in order to infer aerosol size information (e.g., Flagan, 1998). To this end dif-
ferential mobility analysers (DMAs) are commonly used (Knutson and Whitby, 1975). Recent improvements of DMAs made accurate particle sizing down to cluster sizes around 1 nm possible (Brunelli et al., 2009; Steiner et al., 2010; Jiang et al., 2011a; Fernández de la Mora and Kozlowski, 2013). Size-distribution information is then obtained by applying either a step-wise varying (Winklmayr et al., 1991) or continuously ramped voltage (Wang and Flagan, 1990) to a DMA and measuring the downstream aerosol concentration.

In a great number of applications condensation particle counters (CPCs) are used for aerosol detection at single particle counting level. The generation of the supersaturated vapour as well as the used working-fluid itself varies for different CPC types. All of them, either expansion-type with various working fluids (Winkler et al., 2008; Pinterich et al., 2016), mixing-type with diethylene glycol (DEG) (Vanhanen et al., 2011) or the most common laminar-flow-type with butanol (Stolzenburg and McMurry, 1991), water (Hering et al., 2005; Kupc et al., 2013), or DEG (Wimmer et al., 2013) have reached particle detection
at sizes as low as ∼1-2 nm, i.e. close to the critical cluster size.

Particle size-distribution measurements in the sub-10 nm range by combining a DMA with a state-of-the-art CPC are for example reported by Jiang et al. (2011b) and Kuang et al. (2012a). Another approach was used by Lehtipalo et al. (2014) with a particle size magnifier (PSM) operated in scanning mode. There, the instrument's lower size-detection cut-off is varied in order to infer size-distribution information of sub-3 nm particles without relying on the electrical mobility technique.

However, both methods still suffer from a limited time-resolution, as neither the voltage scan at the DMA nor the cut-off scan in the PSM can be performed sufficiently fast. Additionally, diffusional sampling losses get very high for sub-10 nm particles and particle charging gets difficult (Wiedensohler, 1988), therefore signal strength might be weak. In this case a scanning over different sizes with a DMA never allows to exploit the full counting statistics at one size.

On the other hand, the scanning cut-off technique does not rely on particle charging and mobility analysis. But it needs very
careful and broad calibration measurements, as the cut-off strongly depends on the aerosol chemical composition (Kangasluoma et al., 2014). Accordingly, the sizing information might have high systematic uncertainties.

Here we present the development of a newly designed DMA-train setup featuring six DMAs operated in parallel at six distinct but fixed voltages in combination with six state-of-the-art condensation particle counters to obtain fast and precise size-distribution measurements. The idea was first brought forward by Flagan et al. (1991), who showed that especially rapidly
changing aerosol can be tracked with this method. This approach has been used recently to study nanoparticle formation in the NCAR aerosol chamber (Winkler et al., 2013). We now refined it for the application in the cluster-particle transition regime in the sub-10 nm range.



## 2 The DMA-train setup

The DMA-train was planned and constructed as a fixed mounted setup providing all necessary power supplies, flow pumps and controls, including a unified data acquisition system. Furthermore, the inlets to all six channels are kept identical in length and shape to assure equal flow patterns. Figure 1 shows the main design features. In order to keep the instrument as compact as

possible a symmetrical two layer design was chosen, with two identical layers consisting of three DMAs. The inlet is located in between the layers with a common flow for all channels at a flow rate of 11 litre per minute (lpm) maintained by the sample flow of the six used condensation particle counters. In order to reduce diffusional sampling losses core-sampling allows an additional 9 lpm make-up flow in the main sampling line controlled by a critical orifice.

The flow is then split up at a 1/2 inch union-T into two transport flows for the two layers with 5.5 lpm each. The subsequent

design follows the classical approach of a scanning mobility particle sizer (Wang and Flagan, 1990). Each stream is passing through a soft X-ray bipolar Advanced Aerosol Neutralizer (AAN) model 3088 from TSI Inc. Kallinger and Szymanski (2015) found that the AAN reproduces well the predicted charging probability for negative particles predicted by Wiedensohler (1988) for flow rates up to 5 lpm and down to sizes as small as 5 nm. Moreover, Kallinger et al. (2012) reported that the size-distribution of the negative aerosol charger ions does not exceed mobility diameters of 1.6 nm for flow rates even as high as 16

lpm. This basically sets the lower sizing limit of the DMA-train.

After the aerosol has reached a defined charging state at the exit of the AAN, the flow is split up three-fold in a custom-built three-way flow-splitter. Thereby the DMAs receive two 1.5 lpm and one 2.5 lpm streams on each layer of the DMA-train, respectively. The tubing between flow-splitting and the DMA entrance is kept as short as possible.

Afterwards the aerosol is classified in six stainless steel Grimm S-DMAs, which follow the "Vienna-type" design (Reischl,

1991). Some characterization results can be found in Jiang et al. (2011a). The closed-loop sheath-air flow of the DMAs is regulated by temperature controlled critical orifices connected to a single high throughput vacuum pump offering eight separate pumping chambers. The critical orifices of six pumping chambers provide a flow of 15 lpm while two closed loop circuits can be connected to two additional pumping chambers with 10 lpm critical orifices. Accordingly, we can operate 4 DMAs at a sheath flow rate of 15 lpm and 2 DMAs at 25 lpm. The common flow-unit designed by Grimm Aerosol Technik GmbH & Co.

KG is furthermore equipped with sensors for temperature, pressure and relative humidity for the simultaneous monitoring of all eight pumping circuits. The sheath-air is dried by six 2.26 liter volume silica-gel dryers and filtered in active charcoal filters from chemical impurities and in HEPA-filters from remaining aerosol particles. The six identical HV-modules from Grimm Aerosol are able to provide positive voltages up to +6 kV to the central electrode of the S-DMA.

After the classification the detection is performed with six state-of-the-art condensation particle counters. Three channels are

equipped with a TSI 3776 butanol based ultrafine CPC with a cut-off diameter as low as $\sim 2.5$ nm (Stolzenburg and McMurry, 1991; Hermann et al., 2007). For one channel a TSI 3788 water based ultrafine CPC is used (Kupc et al., 2013). The last two channels are operated with a combination of an Airmodus A10 PSM together with a TSI 3776 butanol CPC for the activation of sub-2 nm aerosol (Vanhanen et al., 2011). An additional hyco membrane pump provides the necessary operating vacuum for the Airmodus PSMs. The usage of particle counters with different working fluids introduces a special feature to the DMA-



train: sampling aerosol of the same size in two or three channels with different CPC types can therefore provide information on aerosol composition, as the activation efficiencies of the CPCs depend on the chemical composition of the seed particle (Kangasluoma et al., 2014).

The particle counters are directly connected to the Grimm S-DMA outlet, such that further transport losses are minimized. As the PSMs require a higher sample flow of 2.5 lpm, the two DMAs upstream are operated at a sheath-air flow rate of 25 lpm to keep the DMA resolution similar to the channels with only 1.5 lpm sample flow and 15 lpm sheath flow.

The complete setup is controlled by a National Instruments LabView (Version 2014F) data acquisition software. It provides the control of the HV-modules, the readout of the CPCs and important sheath-air parameters, such that stable operating conditions of all devices can be verified during operation.

## 3   Laboratory characterization

### 3.1   DMA calibration

All six DMAs used in the DMA-train are of identical type and purchased as Grimm S-DMA from Grimm Aerosol Technik GmbH & Co. KG. To test their performance we used a high resolution DMA (UDMA) operated at sheath-air flow rates of $\sim$ 450 lpm, in detail described by Steiner et al. (2010).

For the calibration setup the UDMA was classifying nanoparticles from either a tungsten oxide generator or from an electrospray source to create well defined mobility standards. The calibration setup is shown in Fig. 2 and follows the classical tandem-DMA configuration, which is for example described in detail by Stolzenburg and McMurry (2008). A brief description of the procedure is given in Sect. 3.1.2.

#### 3.1.1   Calibration of the UDMA

Due to the high sheath flow rates in the UDMA, which are not measurable during operation, a calibration of the voltage-mobility relation is necessary. This was done by using clusters of positively electrosprayed tetra-heptyl ammonium bromide (THABr) (Ude and Fernández de la Mora, 2005). The spectrum recorded by a Faraday Cup Electrometer (FCE) downstream of the UDMA shows clear peaks at electrical mobilities of 0.97, 0.65 and 0.53 $\mathrm{cm^2V^{-1}s^{-1}}$, each associated to clusters of the form $\mathrm{A^+(AB)}_n$ of the electrosprayed salt.

A fit to the well identified monomer allows to calibrate the voltage-mobility relation of the UDMA shown in Eq. (1) (Stolzenburg and McMurry, 2008),

$$Z = \frac{1}{V} \cdot \frac{\ln(R_i/R_o) \cdot Q_{\mathrm{sh}}}{2 \cdot \pi \cdot L} \tag{1}$$

by identifying the peak voltage of the monomer with a lognormal fit and then calculating the corresponding sheath air flow rate $Q_{\mathrm{sh}}$. All other parameters of Eq. (1), the outer and inner electrode radii $R_o$ and $R_i$ as well as the length of the classification region $L$ are geometric factors which correspond to the construction values.





### 3.1.2 Calibration of the voltage-mobility relation for six Grimm S-DMAs

In order to calibrate the Grimm S-DMA voltage-mobility relation a tungsten oxide nanoparticle generator from Grimm Aerosol was connected to the UDMA. The calibration of the UDMA done with the positively charged THABr monomer is assumed to be valid as long as the flow conditions are not changed. Tungsten oxide particles of mean diameters 2.0, 2.5, 3.0, 4.0 and 5.0 nm

were classified. The Grimm S-DMA was connected downstream of the UDMA as was the reference FCE (FCE1) measuring the concentration $N_1$ immediately after the UDMA. As shown in Fig. 2 a second electrometer (FCE2) was measuring the concentration $N_2$ downstream of the Grimm S-DMA while the DMA voltage was scanned in logarithmic equidistant intervals. For each classified particle size, a reference measurement was performed with the Grimm S-DMA removed and the FCE2 directly connected to the UDMA, thereby correcting the relative concentration ratio $N_2/N_1$ for nonidealities in flow splitting,

possible different diffusional losses or electrometer offsets. A least-square lognormal fit is used again for identifying the peak voltage. The fitted mean voltages for all selected mobilities and DMAs are plotted in Fig. 3.

To extend the calibration size-range below 2 nm, only negative mobility standards can be used due to the positive high voltages applied to the Grimm S-DMA. In a previous study the ionic liquid Methyl-trioctylammonium bis(trifluoro methylsulfonyl)imide (short MTOA-B3FI) has proven to be a qualified source of negative mobility standards (Steiner et al., 2016). A typ-

ical recorded spectrum downstream of the UDMA while electrospraying MTOA-B3FI is illustrated in Fig. 4. The peaks of the MTOA-B3FI monomer (A$^-$) and dimer (A$^-$(AB)$_1$) are well separated at mobilities of $1.80 \, \mathrm{cm^2V^{-1}s^{-1}}$ and $0.77 \, \mathrm{cm^2V^{-1}s^{-1}}$, respectively. These mobilities correspond to mobility diameters of 1.06 nm for the monomer and 1.62 nm for the dimer. The trimer (A$^-$(AB)$_2$) and the tetramer (A$^-$(AB)$_3$) can be identified in Fig. 4 as well. However, they were not used for calibration measurements as they overlap with the background from multiply charged bigger clusters.

In a second set of experiments the same procedure as for the size-classified tungsten oxide particles was repeated with the UDMA classifying monomer or dimer of the MTOA-B3FI and therefore providing a calibration measurement using a strictly monodisperse aerosol. The peak mobilities retrieved from the lognormal fit are added to Fig. 3. For all calibration runs, the sheath-air flow rate $Q_{sh}$ is measured with a Gilian Gilibrator 2 low-pressure drop bubble flow meter from Sensidyne, LP. For the individual sheath-air circuits of the flow unit the sheath-air flow rate is very stable over time due to the temperature control

of the critical orifices. Therefore, the classification length $L$ is used as the free parameter of the least-square fit to the data in Fig. 3, according to the inverse of Eq. (1). For the other parameters, the design values of $R_i = 0.013$ m and $R_o = 0.020$ m are used.

The results summarized in Table 1 are all slightly higher than the specified classification length by the manufacturer of 13 mm, but are still in reasonable agreement. Moreover, all six DMAs seem to be identical within the uncertainties of the

measurements.

### 3.1.3 Retrieval of the transfer function of the Grimm S-DMA

Additionally, the well defined mobility of the classified MTOA-B3FI monomer and dimer allows to infer information about the DMA transfer function and its penetration characteristics. According to Jiang et al. (2011a), the response downstream of



the test DMA can be written as

$$\frac{N_2}{N_1} = \eta_{\mathrm{dma}}\left(d_p\right) \cdot \Omega\left(V, Z\left(d_p\right)\right) , \tag{2}$$

because only the monodisperse MTOA monomer or dimer particles were sent into the Grimm S-DMA.

Here $N_2$ and $N_1$ correspond to the FCE concentration readings downstream and upstream of the test DMA, $\eta_{\mathrm{dma}}$ represents

a loss parameter for losses in the inlet and outlet regions of the Grimm S-DMA and $\Omega\left(V, Z\left(d_p\right)\right)$ is the DMA transfer function. The notation we use here and details about DMA transfer functions can be found in Stolzenburg and McMurry (2008). In general the transfer function yields the fraction of particles with mobility $Z$ passing the DMA operated at a fixed voltage $V$. However, in the setup presented in Fig. 2 the DMA voltage was not fixed to one value and monodisperse particles of different mobilities were sent into the DMA. In the measurements presented here, the size of the particles was fixed with the UDMA

and the voltage of the tested DMA was scanned. Hence, the measured response cannot be directly projected back into the transfer function as the transfer function of the DMA becomes a function of the DMA voltage due to the significant diffusional broadening below 2 nm (Stolzenburg, 1988; Jiang et al., 2011a).

The transfer function properties are therefore retrieved by a least-square fit of Eq. (2) to the measured ratio $N_2/N_1$ by assuming Stolzenburg's diffusive transfer function (Stolzenburg and McMurry, 2008) including a voltage dependence of the

transfer function width $\sigma_{\mathrm{theo}}(V)$. As suggested by Jiang et al. (2011a), two fit parameters are used: The penetration loss parameter $\eta_{\mathrm{dma}}$, which just reduces the transfer function height due to losses in the inlet and outlet region of the DMA and an additional broadening parameter for the transfer function $f_\sigma$. This is a multiplicative factor to the theoretical width $\sigma_{\mathrm{theo}}$, which accounts for instrument nonidealities like electrode misalignment and distortions of the flow pattern.

A representative measurement of the MTOA-B3FI monomer and dimer is shown in Fig. 5 and the results for all DMAs

are reported in Table 1. Generally we find an additional broadening $f_\sigma$ between 1.01-1.22 for all DMAs and both mobility standards. The measurements of the dimer and monomer for the individual DMAs are very consistent for all DMAs confirming the deviation from the Stolzenburg theory. Moreover, it supports the results of the voltage-mobility calibration, that all six DMAs perform similarly, but differ slightly from the manufacturer values.

The penetrated fractions $\eta_{\mathrm{dma}}$ of the inlet and outlet regions of the Grimm S-DMA derived from the monomer and dimer

measurement can be described by considering them as diffusional losses. As suggested by various authors (Reineking and Porstendörfer, 1986; Karlsson and Martinsson, 2003; Jiang et al., 2011a), the modified Gormley and Kennedy equation (Gormley and Kennedy, 1948; Cheng, 2012)

$$\eta_{\mathrm{pene}}\left(d_p\right) = 0.819 e^{-3.66\mu} + 0.0975 e^{-22.3\mu}$$

$$+ 0.0325 e^{-57.0\mu} + 0.0154 e^{-107.6\mu}$$

$$\text{for } \mu > 0.02 \tag{3}$$

$$\eta_{\mathrm{pene}}\left(d_p\right) = 1.0 - 2.56\mu^{2/3} + 1.2\mu + 0.1767\mu^{4/3}$$

$$\text{for } \mu \leq 0.02$$

$$\tag{4}$$





reproduces well the diameter dependence of such losses with $\mu = \frac{DL_{\text{tube}}\pi}{Q}$ and $D$ the particle diffusivity, $Q$ the volume flow and $L_{\text{tube}}$ the length of the tube, all in SI units.

As the flow through the entrance and exit region of the DMA $Q_{\text{ae}}$ is known, it is appropriate to report an effective diffusional length $L_{\text{pene}}$. It can be used in Eq. (3) in order to extrapolate the DMA penetration to different sizes by setting $\eta_{\text{dma}} = \eta_{\text{pene}}(Q = Q_{\text{ae}}, L_{\text{tube}} = L_{\text{pene}})$, i.e. the DMA losses are represented by diffusional losses through a straight tube.

As summarized in Table 1, we find effective penetration lengths close to 1.6 m for all DMAs. The very good transmission characteristics of the Grimm S-DMA are underlined by the fact that the highly diffusive MTOA-B3FI monomer can still be detected downstream of the DMA and the transmission of the MTOA-B3FI dimer (1.62 nm) is even as high as 8-10 %.

The measurements of the transfer function therefore confirm that the Grimm S-DMA is well suited even for measurements in the sub-2 nm regime, where resolution and transmission normally drop significantly.

## 3.2 Characterization of the condensation particle counters

The performance of the three different types of condensation particle counters used in the DMA-train was tested with the calibration setup shown in Fig. 6. Silver nanoparticles were generated in a tube furnace and then sent into one of the six calibrated Grimm S-DMAs for size classification. The DMA was operated at 1.5 lpm aerosol flow, controlled by the throughput of the silver furnace and at 15 lpm sheath-air flow. An additional make-up flow was supplied after the DMA exit to account for the flow-rates of the detectors. The total flow was then split up at a four-way flow splitter similar to TSI model 3708 with two exits closed.

The counting efficiency is calculated as the ratio of the measured concentrations of the particle counter under investigation and the reference FCE. The results of three typical efficiency measurements are shown in Fig. 7. In order to represent the functional form of the activation curve best, we use the "Gompertz"-function (Gompertz, 1825; Winsor, 1932)

$$\eta_{\text{cpc}}(d_p) = A \cdot e^{\left(-e^{\left(-k \cdot (d_p - d_{\text{p0}})\right)}\right)} \tag{5}$$

and fit it within a least-squares routine to the observed data. Free parameters are the plateau height $A$, the diameter offset $d_{\text{p0}}$ and the curvature parameter $k$.

From the fit a $d_{\text{p50}}$ cut-off diameter is inferred, where the counting efficiency reaches 50 % of the plateau height $A$. The temperature settings we used are reported in Table 2 together with the determined cut-off diameters. We find that the DEG based PSM together with the TSI 3776 for detection achieves the lowest cut-off, well below 2 nm. Due to the rather moderate temperature settings with a $\Delta T = 75\,^{\circ}\text{C}$ between saturator and growth tube the cut-off does not reach the instruments specifications as low as 1 nm. However these settings allow for an operation with very low background from homogeneous nucleation occurring inside the PSM, guaranteeing a high signal-to-noise ratio and still sufficient activation above 1.7 nm.

The two TSI particle counters of type 3776 and 3788 reach the cut-off values specified by the manufacturer. In contrast to the PSM the two counters clearly reach 100 % counting efficiency above ∼4 nm (see Fig. 7). Thus the parameter $A$ of Eq. 5 is set to 1 in the reported fits. The water-based TSI 3788 CPC shows the higher $d_{\text{p50}}$ cut-off due to the unfavourable conditions for



nucleation of water on silver. Much better activation properties for the water-based 3788 are reported when sampling sodium chloride particles (Kangasluoma et al., 2014).

The composition dependence of the $d_{\mathrm{p50}}$ cut-off values of the various condensation particle counters should be kept in mind, when investigating undefined aerosol. In this first approach we can however already verify that our used particle counters

operate as specified by the manufacturers. Moreover, most channels in the DMA-train can be set such that the classified particle diameter lies in the plateau region, well above the cut-off. Thus, inferred particle number concentrations should not depend too much on the pattern of the size-dependent counting efficiency curves.

### 3.3  Instrument transmission

With the DMA-train fully set up, a calibration of the instrument's sampling efficiency was done. Nanoparticles were produced

either in the silver tube furnace ($>5\,\mathrm{nm}$) or in the tungsten oxide particle generator ($<5\,\mathrm{nm}$) and subsequently classified with one of the Grimm S-DMAs removed from the rack. The aerosol flow from the generators was diluted in order to achieve the $20\,\mathrm{lpm}$ sample flow required by the DMA-train. Directly at the main inlet of the DMA-train, an additional core-sampling probe was used for measuring the inlet concentration with an FCE. With all instruments in operation inside the DMA-train a second FCE was installed at the position of the removed DMA used for aerosol size classification and hence measuring the

concentration occurring at the DMA inlet in standard operation. In order to account for FCE measurement offsets, a cross-calibration of the two FCEs was done over the full possible FCE-current measurement range. The concentration ratios of the two FCEs corrected for the offset are shown in Fig. 8.

We present two different types of transmission measurements, in order to account for the different sample flow rates. As expected, the channels with a higher sample flow rate of $2.5\,\mathrm{lpm}$ achieve a higher transmission compared to the channels with

a $1.5\,\mathrm{lpm}$ sample flow rate, which can be seen in Fig. 8. Among channels with the same volume flow rates no significant differences were found.

The functional dependence of the sampling losses is fitted with a threefold Gormley and Kennedy equation. This accounts for the fact that the flow within the sampling procedure is reduced in two steps. After the $20\,\mathrm{lpm}$ main sample flow, the chargers are passed with a flow of $5.5\,\mathrm{lpm}$, followed by the final sampling flow after the three-fold flow splitter. The fitted function is

reported in Eq. (6)

$$
\begin{aligned}
\eta_{\mathrm{sam}} = &\ \eta_{\mathrm{pene}}\left(Q = 20\,\mathrm{lpm}, L = 0.85\,\mathrm{m}\right) \\
&\cdot \eta_{\mathrm{pene}}\left(Q = 5.5\,\mathrm{lpm}, L = L_{\mathrm{eff1}}\right) \\
&\cdot \eta_{\mathrm{pene}}\left(Q = 1.5/2.5\,\mathrm{lpm}, L = L_{\mathrm{eff2}}\right).
\end{aligned}
\tag{6}
$$

As the main sampling line upstream of the core-sampling probe is just a straight tube of $0.85\,\mathrm{m}$ length, this part of the fitting

function is fixed. When the long sampling probe is not used, it can just be neglected. The fit is therefore left with two free parameters $L_{\mathrm{eff1}}$ and $L_{\mathrm{eff2}}$, summarized in Table 3.





## 4 Size-distribution measurements

The fully characterized DMA-train can be used to infer size-distribution information of sub-10 nm aerosol. For data inversion it is preferable to combine the DMA-train data with other instruments covering the size range above 10 nm. This reduces possible systematic errors from multiply charged bigger particles. However, for well-defined chamber aerosol with no particles

bigger than 100 nm, multiple charge effects in the DMA-train can be safely neglected as the bipolar charging probabilities below 10 nm are dominated by singly charged particles.

Raw data are inverted according to the procedure of Stolzenburg and McMurry (2008). We use

$$
\left.\frac{\mathrm{d}N}{\mathrm{d}\ln d_p}\right|_{d_p^*} = \frac{N \cdot a^*}{\beta \cdot f_c(d_p^*) \cdot \eta_{\mathrm{sam}}(d_p^*) \cdot \eta_{\mathrm{cpc}}(d_p^*) \cdot \eta_{\mathrm{dma}}(d_p^*)} \, ,
\tag{7}
$$

assuming symmetrical flow conditions at the DMAs (Jiang et al., 2011b). Here $N$ represents the raw counts of the particle

counter used in the DMA-train channel (PSM channels are corrected for their internal instrument dilution) operated at classifying diameter $d_p^*$. $\beta$ is the aerosol to sheath flow ratio used in the DMAs, $f_c$ corresponds to the Fuchs charging efficiency according to Wiedensohler's approximation (Wiedensohler, 1988) and $a^* = (-\mathrm{d}\ln Z/\mathrm{d}\ln d_p)|_{d_p^*}$. The different diameter dependent efficiencies $\eta$ are already explained in detail in the previous section.

If the channels of the DMA-train are then set to different diameters, an approximation of the measured aerosol size-

distribution can be obtained by linearly interpolating between the six channels. It should be kept in mind, that if the spacing between the different size-channels gets bigger, local structures of the size-distribution cannot be resolved any longer. However for a wide range of measurement applications the inferred size-distribution information will be precise enough to infer the aerosol dynamics in the range below 10 nm. This is demonstrated in the following with two example measurements.

### 4.1 Fast changing aerosol of a tungsten oxide generator at warm-up

In order to test the performance of the DMA-train with an aerosol which undergoes very rapid changes, we connected the tungsten oxide generator directly to the inlet of the DMA-train. For comparison, a standard differential mobility particle spectrometer (DMPS) system (Winklmayr et al., 1991) was connected to the core-sampling probe at the entrance of the DMA-train. The DMPS system uses a FCE as particle detector, which sampled the aerosol at a flow rate of 2.9 lpm. The tungsten oxide generator was then switched on and we followed the evolution of the size-distribution with both systems in parallel, which is

shown in Fig. 9.

The DMPS system only achieves 5 scanning cycles during the eight minutes of warm-up, although its particle scanning range is already reduced to an interval comparable to the DMA-train. As the DMPS is operated such that each scanning cycle starts at the highest voltages, i.e. at the biggest particle sizes, it already detects the first small particles at the end of the second scanning cycle. In Fig. 9 however the scanning information is binned into the time window between start and end of the scan. During

the second scan the generator almost achieved its full performance. No details of the warm-up procedure could therefore be resolved with the classical system.



The DMA-train on the other hand is evaluated at its maximum resolution of 1 s, providing information about the warm-up between 1.7 and 7 nm. It can be clearly seen that as the tungsten oxide coil warms up, first particles of smaller diameter are produced and then, subsequently, the size-distribution reaches bigger sizes.

The absolute concentrations agree reasonably well, thus verifying our calibrations presented in the previous sections.

## 4.2 DMA-train operated at the CLOUD experiment

The DMA-train was operated at the CLOUD experiment during a technical run, where instruments could be tested intensively. CLOUD is described in detail elsewhere (Kirkby et al., 2011; Duplissy et al., 2016) and recently published detailed results about pure biogenic nucleation (Kirkby et al., 2016) and subsequent growth (Tröstl et al., 2016).

Figure 10 shows the time evolution of the different size channels of the DMA-train during a typical nucleation event from pure $\alpha$-pinene ozonolysis in the CLOUD chamber. The ozonolysis was started at time 0 by switching off the electrical field cage inside the chamber and therefore allowing ion-induced nucleation to start (Kirkby et al., 2016). The subsequent growth of the particles could be tracked by the DMA-train. The signal consecutively appears at the different particle size bins, clearly showing the growth of the particles from smaller to bigger sizes. This already allows for the determination of size-dependent growth-rates by relating the signal appearance times to their classified diameters (e.g, Tröstl et al., 2016). In Fig. 10 in total 9 different sizes are covered as some DMAs are set to higher voltages and thus bigger particles, as soon as a steady state concentration is reached in the smaller size channels. This opens an opportunity for covering wider size-ranges with the DMA-train during particle growth events. In this model case of growth driven by organic vapours, the signal rises from 0 to its peak value in one channel within 10 minutes. This clearly points out the need for sufficient time resolution, which is possible with the DMA-train.

On the other hand, CLOUD is typically operated close to atmospheric conditions, and particle number concentrations are low. Hence, the expected signal, especially below 2.5 nm, is only in the range of a few counts per minute. As there is no scanning cycle involved, data is acquired constantly at each size, allowing for averaging over longer time periods in order to even identify very low counts. This can be seen in Fig. 10, where a 2 minute average was applied to the larger size channels, but a 5 minute average to the 1.7 and 2.0 nm channels in order to account for the lower count rates at smaller sizes. Although sampling losses are highest and charging and activation efficiencies are lowest below 2 nm, respectively, the DMA-train can resolve even that early growth, which was not possible by a fast scanning nano-SMPS (Tröstl et al., 2015), connected to the chamber at the same time.

## 5 Conclusions

The DMA-train concept has been presented and its performance was demonstrated by calibration experiments of DMAs, CPCs and sampling losses. These measurements showed that our particle counters operate as specified by the manufacturers. The usage of different CPC types might allow estimates of the chemical composition of the sampled aerosol at different sizes, due to the different activation probabilities of the counters with respect to the seed particle composition. All six Grimm S-DMA





reach accurate size classification with a transmission as high as 8-10 % at diameters as low as 1.6 nm. The sampling losses are minimized as much as possible by using a compact setup, which only needs short sampling tubes and provides high flow rates during the sampling procedure.

Fast changing aerosol can be measured with a time resolution as low as 1 sec. This was demonstrated by the measurement of a tungsten oxide generator during warm-up. The comparison to a DMPS system measuring in parallel showed that the conventional system is not able to track the fast changing aerosol. Furthermore the comparison could as well verify the absolute concentrations inferred from the DMA-train.

The DMA-train is fully mobile and therefore perfectly adjusted for measurements at atmospheric conditions as demonstrated during an intensive measurement campaign at the CLOUD experiment. Measuring at several fixed sizes in parallel offers the possibility of using the full counting statistics. At low concentrations and small sizes an averaging of the signal allows then to lower the sensitivity significantly compared to other state-of-the-art instruments which infer size-distribution information through scanning procedures.

Thus, the DMA-train allows to bridge the gap between measurements in the cluster size-range obtained by mass-spectrometry or scanning PSMs and results from conventional SMPS systems above 10 nm. Furthermore it provides a high-time resolution to even observe very fast aerosol growth, especially in the critical sub-10 nm, where diffusional losses are very high and the survival of freshly nucleated particles is crucial for growing them to atmospherically relevant sizes.

If combined with other measurement devices such as conventional SMPS systems, the full particle size-range can be measured, beginning at cluster sizes as low as 1.6 nm, which is the lower size limit for the DMA-train due to the charger ions. This might allow us to perform detailed comparisons to aerosol growth models (Rao and McMurry, 1989) or apply inverse modelling procedures in order to obtain precise size and time dependent growth-rates (Verheggen and Mozurkewich, 2006; Kuang et al., 2012b). Hence the DMA-train opens a new field for the analysis of aerosol growth processes.

*Author contributions.* D.S. and P.M.W. designed the setup, D.S and G.S performed the calibration experiments, D.S performed the size-distribution measurements, D.S.,G.S and P.M.W. were involved in the scientific interpretation and discussion, D.S.,G.S and P.M.W. wrote the manuscript.

*Acknowledgements.* We thank Andrea Ojdanic for her help with the CPC characterizations and Iris Brodacz for her support in operating the UDMA. We gratefully acknowledge the CLOUD experiment for providing us the opportunity for instrument testing. This work was supported by the European Research Council under the European Community's Seventh Framework Programme (FP7/2007–2013)/ERC grant agreement No. 616075 and the Austrian Science Fund, FWF, project number P27295-N20.



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





## Charging + Classification + Detection

PSM-CPC Combination

Butanol CPC TSI3776

H$_2$O CPC TSI3788

Butanol CPC TSI3776

Butanol CPC TSI3776

PSM-CPC Combination

sample inlet

TSI 3088 Charger

Grimm S-DMA

**Figure 1.** Scheme of the DMA-train setup. The design follows a classical SMPS design in each channel. After bringing the aerosol to a well defined charging state, it is split and classified in six identical Grimm S-DMAs and subsequently detected in condensation particle counters. For optimal detection of sub-2.5 nm particles a combination of a Airmodus A10 particle size magnifier (PSM) with a butanol based TSI 3776 CPC is used. Particles equal or larger than 2.5 nm are detected by either TSI model 3776, or TSI model 3788 CPCs to allow for maximum flexibility with respect to particle activation properties. The whole setup measures 80x135x140 cm, excluding the flow supply unit.




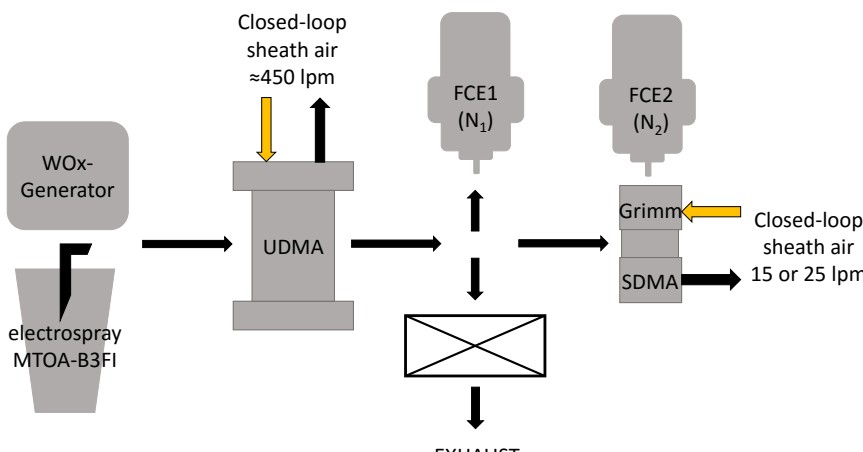

**Figure 2.** Calibration setup for the retrieval of the Grimm S-DMA transfer function and the gauging of the voltage-mobility relation. After aerosol generation particles are classified in a high resolution DMA (UDMA) and then led into the Grimm S-DMA. Two Faraday Cup Electrometers (FCEs) are measuring the concentration upstream and downstream of the Grimm S-DMA.

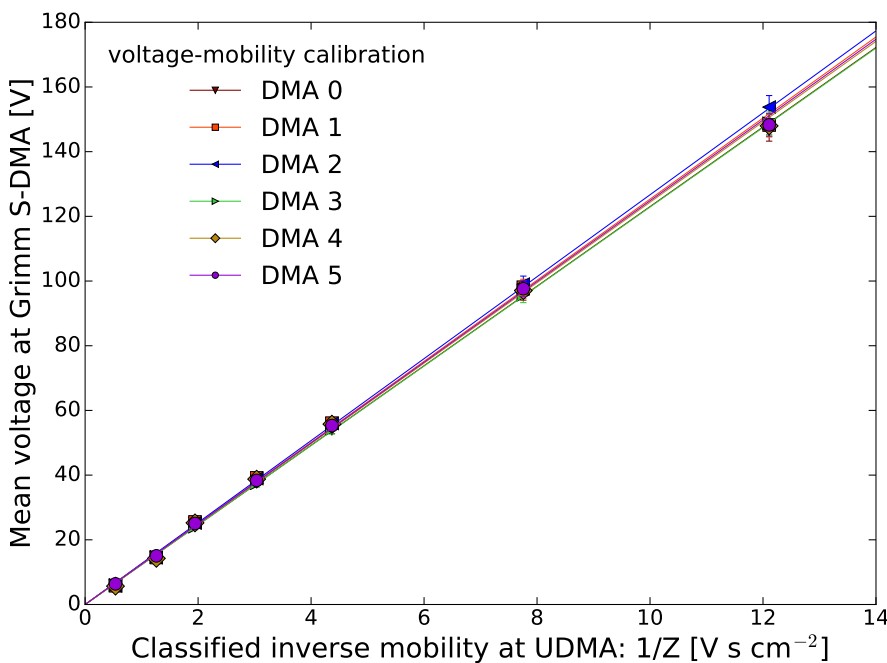

**Figure 3.** Calibration of the voltage mobility relation for all six used Grimm S-DMAs. The mean voltage of least-square fits to the relative FCE response downstream of the six Grimm S-DMA is plotted against the classified inverse mobility of the UDMA. The linear least-square fit has only one free parameter: The effective classification length of the Grimm S-DMA.





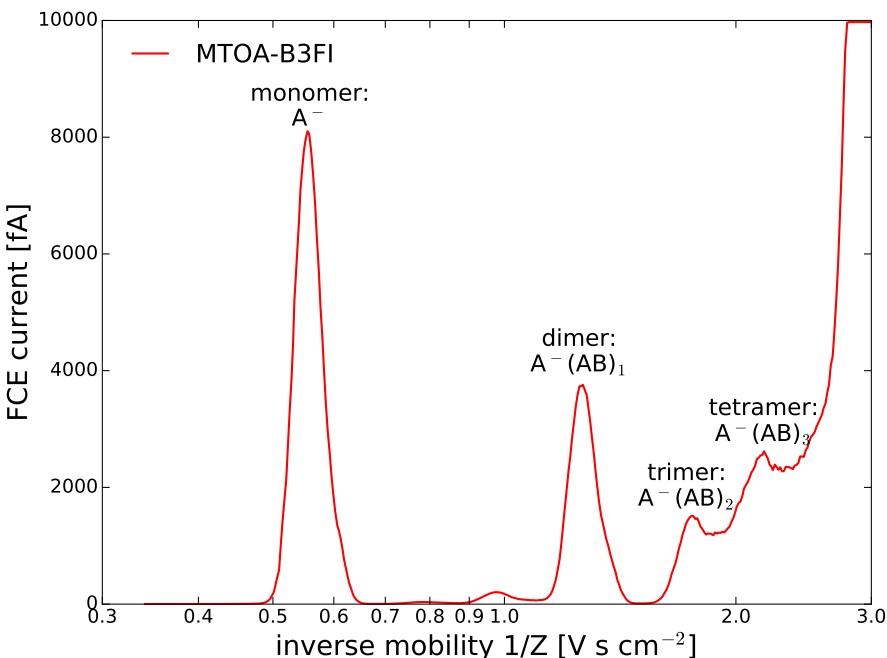

**Figure 4.** Typical MTOA-B3FI spectrum as recorded by a FCE downstream of the UDMA. The peaks of the monomer and dimer are clearly visible and well separated. The trimer and tetramer can be identified as well, but are overlapped by a background of multiply charged larger clusters.





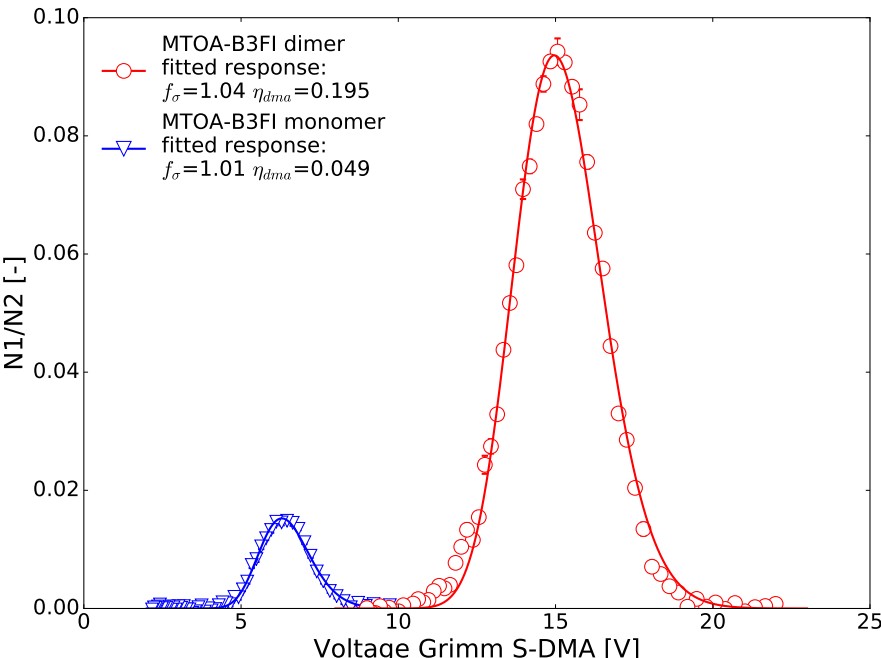

**Figure 5.** Voltage scan of a Grimm S-DMA while the UDMA is classifying either MTOA-B3FI monomer or dimer. The measurements are fitted by the expected response based on Stolzenburg's diffusive transfer function and two additional free parameters: A penetration efficiency $\eta_{\mathrm{dma}}$ and an additional width parameter $f_\sigma$. Reported results correspond to DMA serial number 5.

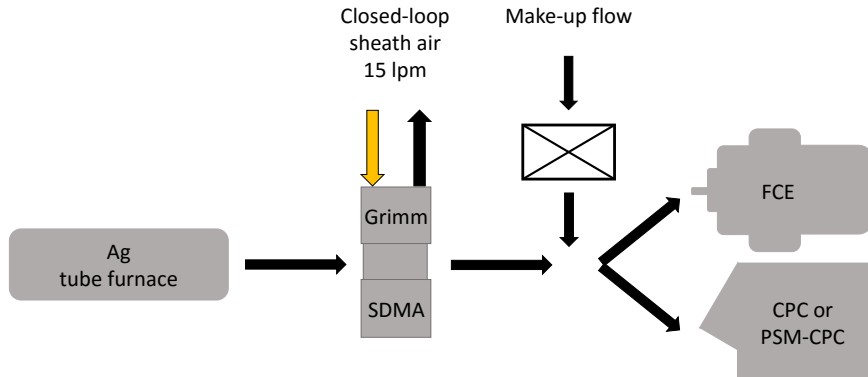

**Figure 6.** Calibration setup for the determination of the counting efficiency curve of the three types of used condensation particle counters. Silver aerosol particles are generated in a tube furnace and classified in one of the Grimm S-DMAs. Afterwards the flow is split into the CPC under investigation and a reference FCE.





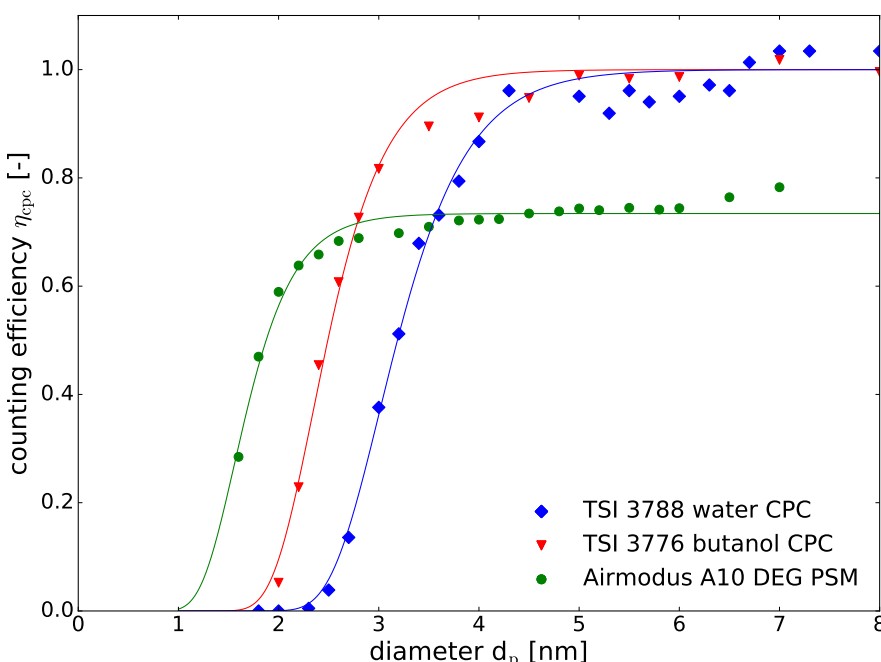

**Figure 7.** Characterization of the counting efficiency with silver nanoparticles for the used three types of condensation particle counters. Green represents the DEG-based Airmodus PSM A10, red the butanol-based TSI 3776 and blue the water-based TSI 3788.





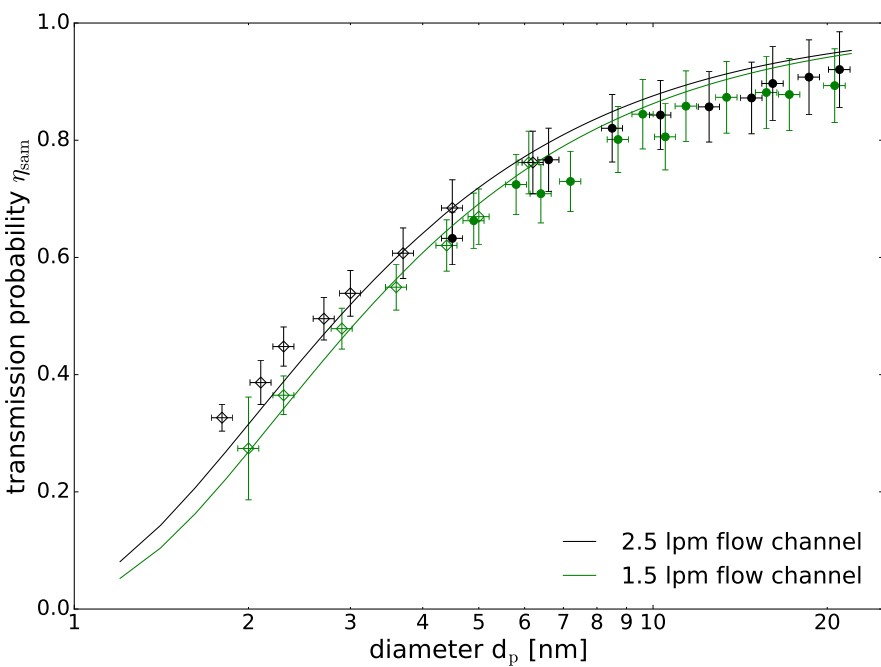

**Figure 8.** Transmission calibration of the DMA-train. The green points show a measurement for the channels operated with a 1.5 lpm final sample flow (the CPC channels) and the black points for the channels with a 2.5 lpm final sample flow (the PSM channels). Open symbols are measured with tungsten oxide aerosol particles and filled symbols are measured with silver aerosol particles. The solid lines represent a threefold Gormley and Kennedy equation fit with two free parameters.





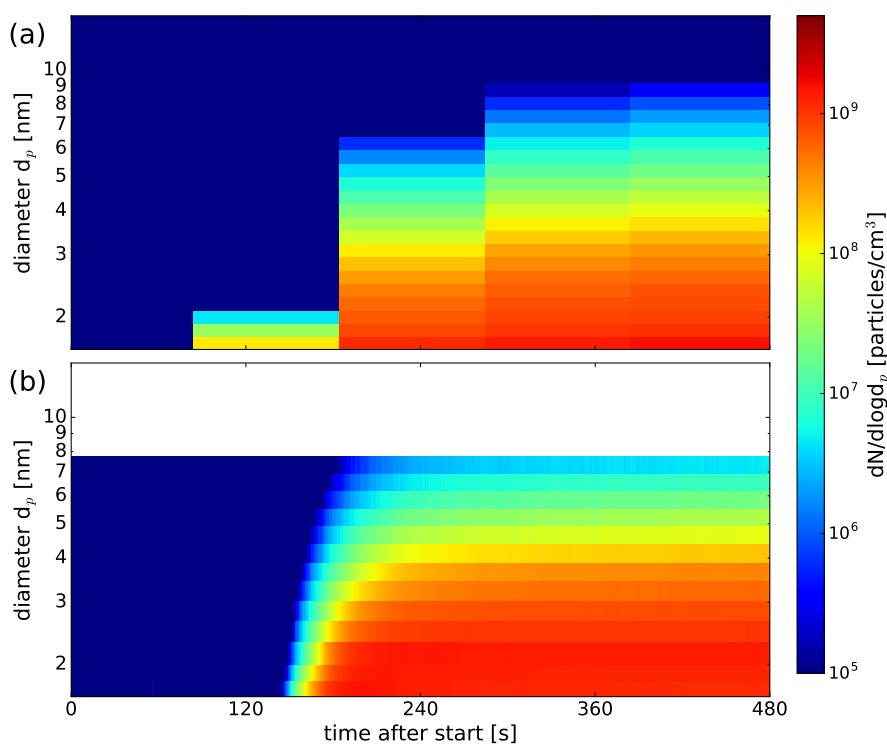

**Figure 9.** Evolution of the size-distribution measured by a conventional DMPS (a) and the DMA-train (b) of a tungsten oxide generator during warm-up. The generator is switched on a time 0. The DMA-train uses its maximum resolution of 1 s while the DMPS bins are placed between start and stop of the scanning cycle.





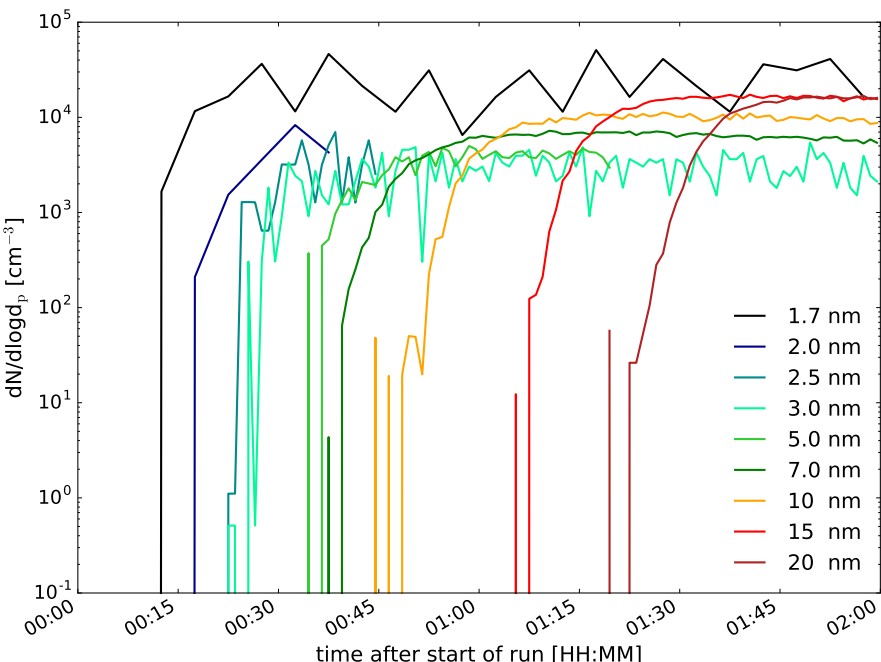

**Figure 10.** Time evolution of the signal in the different DMA-train channels during an $\alpha$-pinene ozonolysis event in the CLOUD chamber. In order to cover a wider size-range, some DMAs are switched to bigger sizes as soon as a steady state is reached. The clear subsequent appearance of the signal in the different channels verifies the size calibration and allows for the determination of particle growth-rates.



**Table 1.** Voltage-mobility calibration and transfer function characteristics of all used Grimm S-DMAs. $L$ is the calibrated classification length of the DMA, $f_\sigma$ accounts for additional transfer function broadening and $L_{\text{pene}}$ represents an effective diffusional length for inlet and outlet losses of the DMA.

| DMA serial number | L [mm] | classified mobility standard | $f_\sigma$ [-] | $\eta_{\text{pene}}$ [-] | $L_{\text{pene}}$ [m] |
|---|---|---|---|---|---|
| 0 | $(13.5 \pm 0.2)$ | MTOA-B3FI monomer | $(1.22 \pm 0.03)$ | $(0.049 \pm 0.001)$ | $(1.59 \pm 0.14)$ |
| | | MTOA-B3FI dimer | $(1.18 \pm 0.01)$ | $(0.178 \pm 0.001)$ | |
| 1 | $(13.3 \pm 0.2)$ | MTOA-B3FI monomer | $(1.12 \pm 0.01)$ | $(0.041 \pm 0.001)$ | $(1.73 \pm 0.14)$ |
| | | MTOA-B3FI dimer | $(1.14 \pm 0.01)$ | $(0.158 \pm 0.017)$ | |
| 2 | $(13.1 \pm 0.2)$ | MTOA-B3FI monomer | $(1.15 \pm 0.03)$ | $(0.037 \pm 0.003)$ | $(1.79 \pm 0.14)$ |
| | | MTOA-B3FI dimer | $(1.11 \pm 0.02)$ | $(0.150 \pm 0.004)$ | |
| 3 | $(13.5 \pm 0.2)$ | MTOA-B3FI monomer | $(1.05 \pm 0.01)$ | $(0.038 \pm 0.003)$ | $(1.59 \pm 0.07)$ |
| | | MTOA-B3FI dimer | $(1.02 \pm 0.01)$ | $(0.182 \pm 0.005)$ | |
| 4 | $(13.4 \pm 0.2)$ | MTOA-B3FI monomer | $(1.20 \pm 0.02)$ | $(0.053 \pm 0.002)$ | $(1.45 \pm 0.09)$ |
| | | MTOA-B3FI dimer | $(1.09 \pm 0.02)$ | $(0.205 \pm 0.032)$ | |
| 5 | $(13.3 \pm 0.2)$ | MTOA-B3FI monomer | $(1.01 \pm 0.02)$ | $(0.049 \pm 0.002)$ | $(1.50 \pm 0.09)$ |
| | | MTOA-B3FI dimer | $(1.04 \pm 0.01)$ | $(0.195 \pm 0.014)$ | |

**Table 2.** Results of the cut-off diameter measurements with Ag seed particles for the three types of used particle counters and the corresponding settings for the measurements. For the TSI instruments we used the standard manufacturer settings.

| CPC type | Settings | $d_{\text{p50}}$ |
|---|---|---|
| TSI 3776 | $T_{\text{sat}}{=}39\,°C$, $T_{\text{con}}{=}10\,°C$ | $(2.5 \pm 0.1)$ |
| TSI 3788 | $T_{\text{con}}{=}15\,°C$, $T_{\text{gt}}{=}75\,°C$ | $(3.2 \pm 0.1)$ |
| Airmodus A 10 | $T_{\text{sat}}{=}80\,°C$, $T_{\text{gt}}{=}5\,°C$, $T_{\text{in}}{=}40\,°C$ | $(1.7 \pm 0.2)$ |

**Table 3.** Results of the transmission calibration of the DMA-train. The two free fit parameters for the threefold Gormley and Kennedy equation (6) are reported for the two sample flow rates.

| Final sample flow | $L_{\text{eff1}}$ [m] | $L_{\text{eff2}}$ [m] |
|---|---|---|
| 2.5 lpm | $(1.6 \pm 0.1)$ | $(0.75 \pm 0.04)$ |
| 1.5 lpm | $(1.6 \pm 0.1)$ | $(0.88 \pm 0.04)$ |