# Peer review of "A DMA-train for precision measurement of sub 10-nm aerosol dynamics"

_Atmospheric Measurement Techniques, 2016_

## Referee Comment (RC1) · Anonymous Referee #1 · 16 Dec 2016

First, I wish to congratulate the authors for putting together such a complicated battery of state-of-the-art aerosol instrumentation, and also for being able to operate it successfully. The instrument certainly is unique in the field of aerosol research and instrumentation, and deserves to be published in AMT. Prior to publication I have couple of general concerns and a few more technical concerns here and there regarding the manuscript, which need to be addressed.

General:

- The authors claim that high time resolution is the advantage of the current DMA train method to study sub 10 nm particle dynamics, and that existing methods are not sufficiently fast to do the same job. However, based on the manuscript it is not clear what is sufficient time resolution and why, to study the sub 10 nm particles. The time resolution of the DMA train itself has not been studied but still the authors make such claims. 1 s mentioned in the text, what is this value based on? Response time is normally measured so that the particle concentration of the sample flow at the instrument inlet rapidly increases from zero to constant value, or similarly decreases from constant value to zero, and from there the 95% values are taken from the CPC readings. If response time of a CPC is ∼1 s (3776 and 3788 can be a little faster), I don't believe that the response time of the whole DMA train is 1 s, at least without shown data, due to the sampling lines and the DMA upstream of the CPCs. I am happy to be proven wrong. Also, the introduction severely lacks references prior literature on fast aerosol sizing methods. Clarity on these need to be significantly improved to give the reader the possibility to put the new instrument into context.

- P8-9, what is missing from the instrument characterization is a figure, in which the x-axis is the particle diameter, and y-axis is the "total transmission" or "total detection efficiency" of the DMA train, i.e. all sampling losses, charging efficiency, DMA transmission efficiency and CPC calibration curves combined. With this the reader can appreciate the performance of the instrument and its suitability in various environments with various particle concentrations. Jiang et al. 2011 (45:510–521, AST), Fig2 is a nice example.

Minor:

- P1 l11-13, which gap? PSM-NAIS-DMPS cover size range from 1 nm to 1 um.

- P2 l1-2, aerosol growth is well studied phenomena, please check for example Ehn et al., 2014 and Tröstl et al. 2016, Nature, and lots of other research on the subject.

- P2 l18-19 and l24-25, not strictly true since the calibration always relies on DMA techniques

- P2 l20-21, not true again at least in the case of DMA, see eg. Tröstl et al. 2015 JAS, or Shah et al. 2005 AST, and certainly some other references

[Figure]

- P2 l20-21, what is sufficient time resolution, and why?

- P2 l21-22, this is the case for the DMA train too

- P2 l24-26, the composition dependency is not due to the supersaturation scanning technique but property of heterogeneous nucleation! It is exactly the same case as in using DMA-CPC system if measuring close to the CPC cutoff diameters, which is always the case below 2.5 nm. The difference is just that the single cutoff of the CPC is uncertain, while in the PSM method the range of cutoffs have uncertainties. This of course provides uncertainty in the DMA-CPC method only in the sizes < 2.5 nm, where particle activation cannot be assumed to be 100% for all compositions

- P8 l5-7, (same as above) exactly, however in the sub 2.5 nm size it cannot be done with DEG, which requires a small discussion somewhere in the manuscript.

- P2 l31, please refrain from citing conference abstracts since they rarely available

- P5 l14, please refrain from citing to unpublished work

- P7 eq5, what is diameter offset?

- P7 l26-29, what is the particle composition the manufacturer uses to calibrate the PSM? As shown in Kangasluoma et al. 2013 (AST), silver shows higher cutoff compared to the other test aerosols, so can the deviation from the manufacturer number be due to different particle chemical composition?

- P10 l14-17, it is mentioned that by following the particle growth, the DMA train can follow the growing mode by changing DMA channels. Is it done automatically or manually? Also, Fig9 shows 13 measured size bins but there are only 6 DMAs. P9 l14-18 mention something about linear interpolation. What are the real measured channels with the DMAs and what are interpolated?

- P10 l2-3, this can be observed from the DMPS data too

- P10 l4, concentration agreement is impossible to judge from the given figure.

- P10 l10, to my understanding switching off the electric field do not start ozonolysis

- P10 l21, how do you know that the counts are not background counts? What is the background count rate of the DMA train, especially with the PSMs?

- P11 l4-7, the experiment did not demonstrate 1 s time resolution

- P11 l11, do you mean increase the sensitivity?
* * *

---

## Author Comment (AC1) · 24 Feb 2017

We appreciate the thoughtful comments by referee #1. For discussion purposes we would like to respond to the general points raised, while the minor points will be addressed in the final response together with an updated version of the manuscript.

We acknowledge the well-justified comment about the time-resolution of the DMA train. Generally, we think that there is a need for fast sizing techniques in the sub-10 nm range, providing both high time-resolution and good counting statistics. The reasons for high (sufficient) time resolution have been pointed out by Wang and Flagan, Aerosol Sci. Technol. 13, 230 (1990). The time resolution needed depends on the actual aerosol system under investigation. For nanoparticles growing at rates between 10 and 100 nm/h scan times around 10 s should be sufficient such that the size distribution

does not undergo significant changes during one scan. However, while fast scanning devices based on electrical mobility analysis are capable of measuring size distribution close to 1 Hz, poor counting statistics in the sub-10 nm size range oftentimes prevents quantitative analysis of nanoparticle dynamics. The fixed size sampling of the DMA train in several channels allows us to monitor particle evolution at a time scale of a few seconds and brings the advantage of using signal averaging at the single sizes and thereby exploiting the full counting statistics.

It is certainly true that the response time of the DMA train is well above 1 second. The transmission of the sampled aerosol through the DMA-train sampling procedure is in the order of 3 seconds based on calculations from the flow velocities. However, due to the symmetry of the six sample channels, we believe that the response time can be accounted for in the data inversion if needed.

The actual time-resolution then depends mainly on the well-known CPC response times. Due to the different counters used, a conservative estimate for the time resolution is therefore rather around 5 seconds. Literature about fast sizing techniques (e.g. Wang et al., 2002; Olfert et al. 2008, Tröstl et al., 2015) will be included in the updated manuscript.

We agree with the referee, that a plot showing both, the overall and the individual contributions to the total detection efficiency similar to Jiang et al. 2011 is worth to be included in the updated manuscript. It is actually readily available as we were already considering to include it in the first version of the manuscript.
* * *

---

## Referee Comment (RC2) · Anonymous Referee #2 · 27 Feb 2017

This manuscript describes a system comprised of a set of six DMAs operated at ~fixed voltage and connected to independent CPCs. The potential advantage of this sort of configuration over a traditional instrument in which a single DMA/CPC is used with stepped or scanned voltage to span a desired particle size range is the improved time resolution that may permit characterization of rapidly evolving aerosols. Careful consideration was given to the component selection and construction of the system. And considerable effort was invested in calibrating and characterizing the DMAs and CPCs. Just that description of the component calibrations alone might justify publication.

The writing would have to be cleaned up a bit before publication, but the paper is certainly readable and understandable in its current form.

My primary concern with the manuscript is simply that for atmospheric research the

time resolution that can be gained from this approach will very rarely justify the expense. Perhaps there are applications in industry for which rapid < 10 nm size distribution changes are common and important. But they aren't identified in the manuscript and the provided justification that this is needed to characterize nucleation mode particles just doesn't seem to be supported by the presented data. The primary example given is of a rapidly evolving size distribution of the aerosol emitted during warm-up of a tungsten oxide generator. But even then the presentation in the paper seems biased to suggest a greater loss of information without parallel measurements. Below I summarize some of my specific concerns. These would need to be addressed before I could recommend publication.

Section 4.1: The manuscript describes measurement of the aerosol emitted during warm up of a tungsten oxide generator. Towards the end of the section the authors state "No details of the warm-up procedure could therefore be resolved with the classical system." Even with the datasets used in the comparison this is an overstatement. Linear interpolation between the size distribution measurements wouldn't capture all of the evolution, but it would certainly do better than provide "no details". Furthermore, the comparison seems designed to provide a large contrast between the two. Specifically, i) a DMPS system was used, rather than a faster SMPS system, and ii) the size range spanned by the DMPS extended out to ∼15 nm while that by the DMA-train extended to only 7 nm. If the argument is that only with a DMA-train can these dynamics be captured then the authors should compare with an SMPS scanning up to 7 nm and with the scan time minimized to the extent possible.

Section 4.2: Unlike the rapid evolution of the generated aerosol, the nucleation mode aerosol sampled in the CLOUD chamber grows rather slowly. The manuscript again argues that an SMPS can't provide the needed time resolution, but this needs to be defended with data. From the text description and Figure 10 it seems the particle growth rate for this example was around 10 nm/hr, which is quite fast. But even then, that is just 0.17 nm/min or about 0.25 nm/DMPS measurement. That sort of change

between measurements should be easily resolved and quantified. I appreciate that low count rate may limit the DMPS stepping or SMPS scanning rate. But this needs to be defended using data and not just asserted to be significant.

Minor issues and comments

Abstract, line 5: Change A 10 to A10

Page 1, line 17: Add "," after (Albrecht, 1989)

Page 2, line 1: Replace "get lost to bigger pre-existing aerosol" with "coagulate with bigger pre-existing particles"

Page 2, line 21: "sufficiently fast" is arbitrary without some reference to the measurement or conditions.

Page 2, line 23: re-word "allows to exploit"

Page 3, line 8: "allows" suggests an option and not necessarily that it was done.

Page 3, line 10: Why does the approach follow an SMPS and not either a DMPS or SMPS?

Page 3, line 29: "state-of-the-art" is subjective. Replace with "modern".

Page 4, line 1: Contrasting the butanol- and water-based CPCs to infer composition is mentioned but never demonstrated (also in the conclusions).

Page 8, line 6: Related to the above comment, the statement that the inferred particle concentrations should not be too dependent on size-dependent counting efficiency also implies that the dependence on working fluid (water or butanol) is also likely pretty modest.

Page 9, line 4: Is the influence of multiply charged particles really relevant for this size range? To keep this statement it should be supported by a back-of-the-envelope estimate of the error introduced by not having an instrument to measure the >10 nm

particles.

Page 9, line 5: The presence of >100 nm particles is irrelevant because they would not show up in the <10 nm size range even if they were multiply charged (with reason). And the bipolar charging probability of <10 nm particles is irrelevant because they are not the particles that would contribute to the signal only because of multiple charges.

Page 11, line 8: "Perfectly adjusted" is subjective.

Page 11, line 10: What does "using the full counting statistics" mean?

Page 11, line 11: Lowering the sensitivity is not a good thing as implied here.

Figure 7: The < 1.0 plateau for the PSM is mentioned in the text, but not explained.